## [Peer Review File · Nature Communications]

MENDER: Fast and scalable tissue structure identification in Spatial Omics DataEditorial Note: This manuscript has been previously reviewed at another journal that is not operating a transparent peer review scheme. This document only contains reviewer comments and rebuttal letters for versions considered at *Nature Communications*.

Reviewer #1 (Remarks to the Author):

In this revision, the results have been expanded to a much wider variety of datasets, and the investigation of parameters has been significantly improved. However, there remain concerns relating to the ways in which MENDER presents a meaningful improvement over prior analysis.

This is a critical point. Without such improvement, the significance and novelty of this paper is not justified, and its publication in this journal is then not warranted.

Specifically, regarding point 1.4: a difference, even if statistically significant, of negligible effect size is not particularly meaningful. The changes in NMI shown in SI Fig. 8, with the median changing on the order of 0.01-0.02, are not large enough to support the claim that using multiple slices meaningfully improves performance. As mentioned in the rebuttal, there are other advantages to combined multi-slice analysis such as the avoidance of separate domain alignment, but the text should be adjusted to make clear that differences are not of a meaningful effect size.

Regarding point 1.7: While there are some scattered red CD8 T-cells inside this region, and a few others around the boundary, the diversity is still very low relative to the orange region, which has not only a diversity of cell types, but also clear spatial variation in the distribution of those cell types, which should create very different cellular contexts (comparing, for example, the bottom left with the center). In conjunction with the previous point, it would be helpful to see a more extensive explanation in the manuscript of how cellular context relates to the determined domain labels.

As a minor point: many SI figure captions do not contain enough information or are repeated over multiple figures. Each caption should clearly indicate what the figure shows and should be specific to that figure.

Reviewer #4 (Remarks to the Author):

I appreciate the authors' efforts in expanding the experimental section in this revised version. The concerns raised about benchmarks have been appropriately addressed. I have a couple of minor suggestions for improvement:

1. The core methodological innovation of MENDER is the use of "multiple ranges" for cellular context representation instead of "single range" in existing methods. While Supplementary Fig. 19 has been included, how the number of ranges and the radius are finally decided is still unclear. Could the authors elaborate on whether the utilization of 4 ranges for slide-seq and 6 for single-cell (as well as the 15um radius) analysis is a general recommendation or if it results from a hyper-parameter search process?

2. I recommend performing a systematic comparison of computational costs between MENDER (both in terms of time and resource usage) and other relevant methods. This comparison would provide valuable insights into the efficiency and practicality of MENDER in comparison to alternative approaches.

Comment 1.0

In this revision, the results have been expanded to a much wider variety of datasets, and the investigation of parameters has been significantly improved. However, there remain concerns relating to the ways in which MENDER presents a meaningful improvement over prior analysis.

This is a critical point. Without such improvement, the significance and novelty of this paper is not justified, and its publication in this journal is then not warranted.

Specifically, regarding point 1.4: a difference, even if statistically significant, of negligible effect size is not particularly meaningful. The changes in NMI shown in SI Fig. 8, with the median changing on the order of 0.01-0.02, are not large enough to support the claim that using multiple slices meaningfully improves performance. As mentioned in the rebuttal, there are other advantages to combined multi-slice analysis such as the avoidance of separate domain alignment, but the text should be adjusted to make clear that differences are not of a meaningful effect size.

Author response:

We greatly appreciate your recognition of the expanded scope of our results and the improvements made in investigating parameters. We understand your concern about whether MENDER presents a meaningful improvement over prior analysis. We would like to address it in detail.

1, MENDER's Improvement Over Prior Analysis:

- Accuracy: MENDER significantly enhances accuracy compared to prior analysis methods. We present extensive evidence in support of this claim. Figures 2, 3, and our new Supplementary Figures 10, 11, 12, and 13 are dedicated to demonstrating the substantial improvements in accuracy achieved by MENDER.
- Running Time Efficiency: Another notable advantage of MENDER is its improved running time efficiency. In Figure 4J, we have provided a direct comparison between MENDER and prior methods, clearly showcasing a remarkable improvement in speed.
- Scalability: MENDER exhibits superior scalability, as depicted in Figures 4A-G. We demonstrate its capability to handle large-scale datasets that were previously beyond the ability of existing methods. This scalability positions MENDER as a unique and valuable tool in the field.

These three points collectively illustrate that MENDER offers substantial improvements over prior analysis methods, setting it apart as an innovative and effective approach.

2, Clarification on SI Figure 8:

We acknowledge your concern regarding SI Figure 8, where the differences in NMI

between the multi-slice version of MENDER (MENDER_multi) and the single-slice version (MENDER_single) appear relatively small. However, it's important to note that both MENDER_multi and MENDER_single are integral components of MENDER itself; they represent the multi-slice and single-slice analysis modules of our proposed method. Therefore, the small improvement of MENDER_multi over MENDER_single does not suggest a lack of improvement over prior analysis. Instead, it highlights the intrinsic benefit of MENDER's multi-slice capabilities within the context of our method.

3. Text Adjustment per Your Suggestion:

In response to your valuable feedback, we have made adjustments to the main text on Page 8 to emphasize the advantages of multi-slice analysis while acknowledging the relatively modest per-slice improvement, as follows:

"The benefit of multi-slice analysis over single-slice analysis, as emphasized in recent studies (Li & Zhou, 2022; Yuan et al., 2022), lies in its ability to perform spatial domain identifications across multiple slices simultaneously. This facilitates the comparison of identified results across slices, avoiding the need for separate domain alignment and reducing the risk of inconsistent clustering granularity across slices. We also assessed whether multi-slice analysis offers per-slice improvement. To do so, we conducted single-slice analysis for each slice across three datasets, resulting in a total of 37 single-slice analyses. We then compared the accuracy (in terms of NMI) of single-slice and multi-slice analyses for each slice (see Methods). While statistically significant, the per-slice improvement of multi-slice over single-slice analysis is relatively small (Supplementary Fig. 8)."

We believe these adjustments clarify the context and significance of multi-slice analysis within MENDER.

Once again, we appreciate your insightful comments and hope that these clarifications address your concerns effectively.

Comment 1.1

Regarding point 1.7: While there are some scattered red CD8 T-cells inside this region, and a few others around the boundary, the diversity is still very low relative to the orange region, which has not only a diversity of cell types, but also clear spatial variation in the distribution of those cell types, which should create very different cellular contexts (comparing, for example, the bottom left with the center). In conjunction with the previous point, it would be helpful to see a more extensive explanation in the manuscript of how cellular context relates to the determined domain labels.

Author response:

Thank you for your insightful comment regarding Point 1.7. We appreciate your careful examination of our work and your valid point regarding the diversity of cellular contexts and its relationship to determined domain labels. We have thoroughly considered your feedback and have taken steps to provide a more comprehensive explanation in our manuscript.

We employed the following additional analyses for the diversity of cellular contexts in different regions:

1. Visualization of Cellular Context: We assessed the variation in cellular context within both regions, specifically, the region with high cell type diversity (namely, R_h) and the region with low cell type diversity (R_l) (Fig. R1B). To achieve this, we used UMAP dimensional reduction to reduce the high-dimensional cellular context representation obtained by MENDER to three dimensions. Additionally, we assigned each cell a color based on its associated cellular context's 3D embedding in the CIELAB color space. This approach of visualizing spatially resolved data has been adopted in previous studies (Parigi et al., 2022; Shang & Zhou, 2022; Solorzano, Partel, & Wählby, 2020). We have included the results in Figure R1D, in which variations in colors reflect the variations in cellular context. This visualization effectively illustrates the cellular context variations within both R_h and R_l regions. In R_h , we observed the presence of dark red, dark green, and blue colors, which correspond to spatial variations in cell types across this region. Similarly, in R_l , we identified variations represented by grass green, light purple, and pink colors, indicating the existence of diverse cellular contexts within this region.

2. Influence of Clustering Resolution: Motivated by your point, we investigated the impact of increasing the Leiden clustering resolution (note that this Leiden clustering is performed on the cellular context representation to obtain spatial domain labels). Our findings supported your hypothesis that R_h should contain different cellular contexts. By increasing the resolution, we observed that the R_l region generated different spatial domain labels (Fig. R1F). Conversely, when we decreased the Leiden resolution, we noted that the domain labels within R_l became more homogeneous (Fig. R1E).

Based on our analysis, we draw the following conclusions:

1. Variations in cellular context play a significant role in the determination of domain labels. Our analysis demonstrates that these variations can result in distinct domain labels.

2. Modifying the Leiden clustering resolution allows for control over the granularity of domain labels, providing a more fine-grained or homogeneous representation of cellular contexts.

Figure R1.
A: Domain label.
B: Cell type label.
C: The same with (B), but excluding cells with homogeneous domain regions, i.e., orange domains in (A). This same triple negative breast cancer data in Figure 6.
D: The color visualization is obtained by: (1) Use UMAP dimensional reduction to reduce the high-dimensional cellular context representation obtained by MENDER to three dimensions. (2) Assign each cell a color by linearly mapping its associated cellular context's 3D embedding to the CIELAB color space.
E: domain labels obtained by decreased Leiden clustering resolution.
F: domain labels obtained by increased Leiden clustering resolution

We added explanation in the manuscript of how cellular context relates to the determined domain labels as follows:

There are primarily two factors that can influence the determination of spatial domain labels. The first factor is cellular context because MENDER relies on the representation of cellular context to determine spatial domain labels. However, it's important to note that the presence of the same spatial domains doesn't necessarily imply the absence of cellular context variations. For instance, consider the original spatial domain region in Figure 6B, which can still contain cellular context variations, as demonstrated by the color variations in the R_h region in Supplementary Fig. 35A. Here, we used UMAP-reduced cellular context representation and mapped it to the CIELAB color space for each cell to illustrate these variations. The second factor is the Leiden clustering resolution. When we increased the clustering resolution, we observed that the R_l region generated different spatial domain labels (Supplementary Fig. 35C). Conversely, when we decreased the Leiden resolution, we noticed that the domain labels within R_l became more homogeneous (Supplementary Fig. 35B).

Comment 1.3

As a minor point: many SI figure captions do not contain enough information or are repeated over multiple figures. Each caption should clearly indicate what the figure shows and should be specific to that figure.

Author response:

Thank you for your feedback. We have carefully reviewed the captions in the Supplementary Information (SI) figures and identified areas for improvement. Specifically, we noticed that in some cases, we had used captions like 'Similar to Supplementary Figure 2 but using the BaristaSeq dataset' for figures such as SI Figure 3. We acknowledge that these captions were not sufficiently self-contained. Similar issues may arise when readers refer to SI Figure 4, 13, and 15. To address this, we have added more detailed information to these captions.

In the case of SI Figure 19 and SI Figure 20, we acknowledge that their captions were quite similar. However, it's important to clarify that SI Figure 19 presents each slice separately, whereas SI Figure 20 provides a summary for all slices across the three datasets.

For SI Figure 23-30, we initially used identical captions for all eight figures because they originated from eight different slices within a single dataset. To enhance specificity, we have now included unique slice identifiers in each figure's caption. A similar approach has been applied to SI Figure 32-34, where we have incorporated tissue-specific information into the captions

Comment 2.0 (General)

I appreciate the authors' efforts in expanding the experimental section in this revised version. The concerns raised about benchmarks have been appropriately addressed. I have a couple of minor suggestions for improvement:

Author response:

Thank you very much for your constructive comments. We have carefully considered all your concerns and responded point-by-point.

Comment 2.1 (Major)

1. The core methodological innovation of MENDER is the use of "multiple ranges" for cellular context representation instead of "single range" in existing methods. While Supplementary Fig. 19 has been included, how the number of ranges and the radius are finally decided is still unclear. Could the authors elaborate on whether the utilization of 4 ranges for slide-seq and 6 for single-cell (as well as the 15 μ m radius) analysis is a general recommendation or if it results from a hyper-parameter search process?

Author response:

We appreciate the your interest in understanding the process behind selecting the number of ranges and the radius for MENDER's cellular context representation. This is indeed a critical aspect of our method's innovation.

The decision to employ 4 ranges for Slide-seq data and 6 ranges for single-cell data, along with a 15 μ m radius, was made following a combination of heuristic reasoning and experimentation. Initially, considering that Slide-seq data approximates or slightly exceeds the typical size of individual single cells (around 10 μ m), we initially reasoned that the recommended ranges should be set lower than in the single-cell case.

During the course of our extensive experiments and validations, we did explore various configurations of parameters, including different ranges and radii and observed that alternative configurations also yielded highly effective results (as shown in SI Figure 19-34). Therefore, we wish to clarify that these recommended parameters are not rigidly fixed but rather derived from extensive testing across multiple datasets. When reporting accuracy in our manuscript, we consistently employed these recommended parameters for consistency and clarity.

We hope this explanation clarifies our approach to parameter selection in MENDER. Thank you again for your valuable comment!

Comment 2.2

2. I recommend performing a systematic comparison of computational costs between MENDER (both in terms of time and resource usage) and other relevant methods. This comparison would provide valuable insights into the efficiency and practicality of MENDER in comparison to alternative approaches.

Author response:

Thank you for the suggestion! The running time comparison was presented in Figure 4J. For memory usage, we compared MENDER with other competing methods in our manuscript (Supplementary Table 1).

Per your suggestion, we added the following text in the revised manuscript.

We conducted a memory usage comparison between MENDER and other competing methods (Supplementary Table 1). We recorded the peak memory usage for each method on every dataset. The results indicated that SingleRange, CNC, and MENDER exhibit the best memory efficiency, as they only require the maintenance of one fixed spatial graph and context representation in memory. It's worth noting that even on the MERSCOPE dataset with over 700,000 cells, MENDER only requires 25 minutes and 80GB+ of memory, showcasing its potential capability to handle datasets of million-level scale.

Computing memory usage (MiB)

Dataset	# Cells	STAGATE	BASS	SOTIP	SingleRange	CNC	MENDER
STARmap	3,190	2,149	1,236	2,865	755	1,027	873
BaristaSeq	11,426	3,663	1,568	3,462	921	1,352	956
MERFISH	378,918	82,766	N/A	N/A	38,655	26,798	56,537
MERSCOPE	734,696	N/A	N/A	N/A	53,541	N/A	86,635

Supplementary Table 1: Computing memory usage of all methods. For each method on each dataset, peak memory usage is recorded. The computing memory usage were examined for all the real data applications.

References

- Parigi, S. M., Larsson, L., Das, S., Ramirez Flores, R. O., Frede, A., Tripathi, K. P., . . . Villablanca, E. J. (2022). The spatial transcriptomic landscape of the healing mouse intestine following damage. *Nature Communications*, *13*(1). doi:10.1038/s41467-022-28497-0
- Shang, L., & Zhou, X. (2022). Spatially aware dimension reduction for spatial transcriptomics. *Nature Communications*, *13*(1), 7203. doi:10.1038/s41467-022-34879-1
- Solorzano, L., Partel, G., & Wählby, C. (2020). TissUUm maps: Interactive visualization of large-scale spatial gene expression and tissue morphology data. *Bioinformatics*, *36*(15), 4363-4365.

Reviewer #1 (Remarks to the Author):

In this revision, the author has clarified the point on the significance of MENDER, and adjusted the manuscript to clearly distinguish between statistical significance and effect size.

However, regarding point 1.1, the new visualizations in SI Fig. 35 shed light on the way in which MENDER appears to be processing this dataset but still do not fully resolve the fundamental concern regarding the counterintuitive domain groupings. Specifically:

1. In SI Fig. 35B, the low resolution clustering looks intuitively reasonable given the dataset. However, the medium (Fig. 6B) and high resolution (SI Fig. 35C) look all look significantly different. Similarly, comparing SI Fig. 12 to SI Fig. 31 shows similarly high levels of variation in terms of number of regions. Choosing resolution/number of regions when not known a priori is always a challenge with Leiden clustering, and this seems to suggest that MENDER still needs a more robust way to do so, as the MENDER output seems highly sensitive to the resolution. The manuscript should include a principled way to choose a resolution and demonstrate that this produces reasonable results.
2. Judging from SI. Fig 35A it seems that the MENDER difference between a cellular niche of "all green" and "all green except 1/a few red" is greater than the variation in MENDER difference in the R_h region, even when the cellular niches vary greatly (e.g. pink vs blue in SI Fig. 35C). The manuscript should make explicit how the neighborhood graph that is fed into Leiden is computed, and should be able to show that greater similarity in the graph/in the MENDER UMAP actually corresponds to biologically similar niches.

Additional minor comments:

3. One of the R_l's in SI Fig. 35A seems misplaced.
4. SI Figs 2-4 (and check others): typo in "Louvian" (should be "Louvain")
5. SI Fig. 9: the number of communities/resolution value should be included as a parameter. Also, the "data" column is not really relevant outside of the review process so should probably be removed.
6. Is the dataset from Fig. 6 in SI Fig. 9?

Reviewer #4 (Remarks to the Author):

I appreciate the authors' efforts in expanding the experiments. I have no further questions.

Comment 1.0

In this revision, the author has clarified the point on the significance of MENDER, and adjusted the manuscript to clearly distinguish between statistical significance and effect size.

However, regarding point 1.1, the new visualizations in SI Fig. 35 shed light on the way in which MENDER appears to be processing this dataset but still do not fully resolve the fundamental concern regarding the counterintuitive domain groupings. Specifically:

Author response:

We sincerely appreciate your positive feedback regarding the enhancements made to our manuscript. In response to your remaining queries, we have provided a detailed point-by-point explanation below.

Comment 1.1

In SI Fig. 35B, the low resolution clustering looks intuitively reasonable given the dataset. However, the medium (Fig. 6B) and high resolution (SI Fig. 35C) look all look significantly different. Similarly, comparing SI Fig. 12 to SI Fig. 31 shows similarly high levels of variation in terms of number of regions. Choosing resolution/number of regions when not known a priori is always a challenge with Leiden clustering, and this seems to suggest that MENDER still needs a more robust way to do so, as the MENDER output seems highly sensitive to the resolution. The manuscript should include a principled way to choose a resolution and demonstrate that this produces reasonable results.

Author response:

We appreciate your insightful comment regarding the critical issue of resolution selection in clustering. Here is what we provided in MENDER regarding this issue.

MENDER addressed the resolution selection issue by introducing an approach to iteratively search the optimal Leiden resolution, based on the expected number of regions. We have incorporated the "res_search" function into MENDER's codebase (available at `utils.py`). This function takes various parameters as input, including "adata" (the spatial data for clustering), "target_k" (the expected number of regions), "res_start" (the initial clustering resolution), "res_step" (the search step size), "res_epochs" (the maximum search epochs), and "random_state" (the random seed). For enhanced clarity, we have included an illustrative depiction of the "res_search" function in Supplementary Figure 36 (the following figure).

We acknowledge that in situations involving unknown biological systems, determining the expected number of regions can indeed be a formidable challenge, one that has not been comprehensively addressed by existing spatial clustering methods. This challenge is not isolated to single-cell/spatial clustering but extends to the broader field of clustering in general. In light of these complexities, we recommend that users explore a range of potential values for the number of regions/resolutions when confronted with such uncertainty. This empirical approach allows for a thorough exploration of the tissue structures.

To validate the efficacy of our approach, we conducted experiments using a large-scale brain dataset obtained from MERSCOPE (information available in Supplementary Fig. 9). When using the default clustering resolution, MENDER successfully identifies fine brain structures, including different cortex layers (CTX L1-L6), Caudate putamen (CP), Cortical subplate (Ctx_sp), Olfactory region (OLF), Pallidus (PAL), Fiber tracts (Fiber_tracts), Ventricular systems (VS), and Lateral septal complex (LSX) (Supplementary Figure 37A). When we set the expected number of regions to 5 using the “res_search” function, MENDER accurately identifies 5 broad brain regions, including BS (Brain stem), CNU (Cerebral nuclei), CTX (Cortex), FT (Fiber tracts), and VS (Ventricular systems) (Supplementary Figure 37B), aligning with the major brain regions defined in the Allen Brain Atlas <http://atlas.brain-map.org/atlas?atlas=1&plate=100960324#atlas=1&plate=100960324&resolution=18.60&x=5227.555338541667&y=3791.999782986111&zoom=-4&structure=343>.

Supplementary Figure 37

We have added the following text in the manuscript:

Determine the optimal clustering resolution. The challenge of determining the appropriate resolution or number of regions in spatial clustering is a common hurdle in the field. To address this challenge, we introduced the "res_search" method in MENDER. This approach enables users to iteratively search for the optimal Leiden resolution, given the expected number of regions (Supplementary Fig. 36). To demonstrate, Supplementary Figure 37 highlights the effectiveness of the "res_search" method in resolution selection. Using a MERSCOPE brain dataset, we showed that MENDER, with default resolution settings, identifies fine-grained structures. However, when applying "res_search" with an expected number of regions set to 5, MENDER accurately discerns broader brain regions, aligning with the Allen Brain Atlas.

Comment 1.2

Judging from SI Fig 35A it seems that the MENDER difference between a cellular niche of “all green” and “all green except 1/a few red” is greater than the variation in MENDER difference in the R_h region, even when the cellular niches vary greatly (e.g. pink vs blue in SI Fig. 35C). The manuscript should make explicit how the neighborhood graph that is fed into Leiden is computed, and should be able to show that greater similarity in the graph/in the MENDER UMAP actually corresponds to biologically similar niches.

Author response:

We appreciate your thoughtful comments and suggestions, and we have taken them into consideration in the following. We summarized your question as two parts: (1) How the neighborhood graph that is fed into Leiden is computed? (2) Demonstrate greater similarity in the MENDER UMAP actually corresponds to biologically similar niches.

First, to address the question of how the neighborhood graph used in Leiden clustering is computed, we have included the following text in the revised manuscript:

MENDER-UMAP Visualization and MENDER Spatial Domains Computation.

Utilizing the "Multi-range Neighborhood Representation" of each cell, dimension reduction and clustering are executed to generate the "MENDER-UMAP Visualization" and the "MENDER Spatial Domains". To create the "MENDER-UMAP Visualization", neighborhood graph (*scanpy.pp.neighbors*) is constructed on the normalized and PCA reduced "Multi-range Neighborhood Representation" (implemented by *scanpy.pp.normalize_total* then *scanpy.pp.log1p* and *scanpy.pp.pca*). Then UMAP (implemented by *scanpy.tool.umap*) is applied on the neighborhood graph. To generate the "MENDER Spatial Domains", the Leiden clustering is employed to cluster on the neighborhood graph (same as before). The clustering resolution of Leiden is set in the following manner: if the expected number of domains is known, a function is implemented to automatically estimate the suitable Leiden resolution. This can be accomplished by executing *run_clustering_normal* (with a positive value as the parameter, for the expected number of domains). Conversely, if the expected number of domains is not available (as in exploratory studies), the Leiden resolution defaults to 0.5. This is achieved by running *run_clustering_normal* (with a negative value for clustering resolution).

As suggested, to show that “greater similarity in the MENDER UMAP actually corresponds to biologically similar niches”, we need to show on a well-studied biological system where “biologically similar” is known. To this end, we answered this question using brain data in Figure 4.

First, It is known that the functions and gene expression naturally formed a gradient pattern along the cortex axis from superficial layer to deep layer (Maynard et al., 2021;

Munoz-Castaneda et al., 2021; Zhang et al., 2021). We asked if MENDER UMAP can reconstruct the biological order of cortex layers. The MENDER UMAP of the whole dataset is shown in Fig. 4H, and the cortex layers are highlighted in Supplementary Fig. 38A. We can see that in MENDER UMAP, the order in MENDER UMAP aligns well with the biological order of cortex layers (Supplementary Fig. 38B).

Supplementary Figure 38

Second, in the MENDER UMAP, we observe that domains belonging to the same brain structure are close together. This is evident in the proximity of cortical regions and hippocampal regions in the UMAP space.

Third, even when brain structures are morphologically distinct across different brain positions, the MENDER UMAP can map them together. For instance, in Figure 4E, the Hippocampal region in Position 2 is significantly different from that in Position 1 in terms of tissue morphology, yet the MENDER UMAP successfully clusters them together (Fig. 4H(4)).

These observations collectively demonstrate that the structures in the MENDER UMAP reflect the structural similarities in biology, reinforcing the correspondence between graph similarity and biological similarity.

Comment 1.3

One of the R_I's in SI Fig. 35A seems misplaced.

>>>Thanks! We have made suggested changes.

SI Figs 2-4 (and check others): typo in "Louvian" (should be "Louvain")

>>>Thanks! We have made suggested changes.

SI Fig. 9: the number of communities/resolution value should be included as a parameter. Also, the "data" column is not really relevant outside of the review process so should probably be removed.

>>>Thanks! We have added the parameter information and removed the "data" column as suggested. We have also made changes in the figure captions.

Is the dataset from Fig. 6 in SI Fig. 9?

>>>The information of dataset from Fig. 6 can be located in Fig. 6A. We didn't include it in SI Fig.9 since it does not pertain to benchmarking purposes.

References

- Maynard, K. R., Collado-Torres, L., Weber, L. M., Uytingco, C., Barry, B. K., Williams, S. R., . . . Jaffe, A. E. (2021). Transcriptome-scale spatial gene expression in the human dorsolateral prefrontal cortex. *Nature Neuroscience*, *24*(3), 425-436. doi:10.1038/s41593-020-00787-0
- Munoz-Castaneda, R., Zingg, B., Matho, K. S., Chen, X., Wang, Q., Foster, N. N., . . . Dong, H. W. (2021). Cellular anatomy of the mouse primary motor cortex. *Nature*, *598*(7879), 159-166. doi:10.1038/s41586-021-03970-w
- Zhang, M., Eichhorn, S. W., Zingg, B., Yao, Z., Cotter, K., Zeng, H., . . . Zhuang, X. (2021). Spatially resolved cell atlas of the mouse primary motor cortex by MERFISH. *Nature*, *598*(7879), 137-143. doi:10.1038/s41586-021-03705-x

Reviewer #1 (Remarks to the Author):

In this revision the author has provided additional clarification regarding the previously raised major points, as well as edits to address the minor points. At this point we have a few remaining minor points to finish preparing the manuscript for submission.

Minor Points

1. There remain a few aspects of the text that explicitly refer to the review process, such as in the data and code availability sections. The author should ensure that review-process specific details are not included in the final manuscript.
2. The discussion section should be updated to include discussion of how MENDER's approach compares and relates to other recent methods for finding heterogeneous regions such as Nest (<https://www.nature.com/articles/s41467-023-42343-x>)
3. If possible, it would be helpful to provide access both via Github (or equivalent). Since your code is already structured as a python package, it would probably not be difficult to upload it to pip, and this is also highly encouraged if not already planned.
4. The level of English is entirely acceptable for reading and understanding the manuscript. However, there is room for proofreading to improve the polish/presentation. A few specific points:
 - a. "Whether support multi-slice analysis" and "Whether output cell context representation" should be corrected for grammar. A suggestion is "Support for multi-slice analysis" and "Availability of cell context representation"
 - b. It would generally be preferable to avoid use of contractions.

Comment 1.0

In this revision the author has provided additional clarification regarding the previously raised major points, as well as edits to address the minor points. At this point we have a few remaining minor points to finish preparing the manuscript for submission.

Author response:

Thank you for all your previous comments, which have improved our work very much! Your remaining minor comments are also very helpful to align our work with most recent papers.

Comment 1.1

There remain a few aspects of the text that explicitly refer to the review process, such as in the data and code availability sections. The author should ensure that review-process specific details are not included in the final manuscript.

Author response:

We have updated as follows.

Data availability

We provide the benchmark datasets as h5ad format, please find how to load them in the tutorial [<https://mender-tutorial.readthedocs.io/en/latest/>]. The MERSCOPE dataset is too large to be uploaded and can be requested at [<https://info.MERSCOPE.com/mouse-brain-data>].

Code availability

The Python implementation of MENDER is available at [<https://github.com/yuanzhiyuan/SOMENDER>]. A tutorial on MENDER package is also available at [<https://mender-tutorial.readthedocs.io/en/latest/>].

Comment 1.2

The discussion section should be updated to include discussion of how MENDER's approach compares and relates to other recent methods for finding heterogeneous regions such as Nest (<https://www.nature.com/articles/s41467-023-42343-x>)

Author response:

Thanks for your suggestion! We also noticed Nest when it was firstly public in NC, providing a new solution for identifying hierarchical tissue structures. We have updated our discussion as follows.

Recent innovations have also focused on identifying the hierarchical tissue structures, enabling region-within-region structure delineation, especially in complex disease⁴⁹.

Comment 1.3

If possible, it would be helpful to provide access both via Github (or equivalent). Since your code is already structured as a python package, it would probably not be difficult to upload it to pip, and this is also highly encouraged if not already planned.

Author response:

Thank you for your suggestion! We have uploaded it to pip, so user can install directly by running “pip install SOMENDER”. We also added this information in our github repo: <https://github.com/yuanzhiyuan/SOMENDER>.

Comment 1.4

The level of English is entirely acceptable for reading and understanding the manuscript. However, there is room for proofreading to improve the polish/presentation. A few specific points:

- a. "Whether support multi-slice analysis" and "Whether output cell context representation" should be corrected for grammar. A suggestion is "Support for multi-slice analysis" and "Availability of cell context representation"*
- b. It would generally be preferable to avoid use of contractions.*

Author response:

Thank you for pointing these out! We have thoroughly checked the manuscript and improved the readability of the paper. Thank you again for all your careful reading!